# Association between Health Literacy and Subgroups of Health Risk Behaviors among Chinese Adolescents in Six Cities: A Study Using Regression Mixture Modeling

**DOI:** 10.3390/ijerph16193680

**Published:** 2019-09-30

**Authors:** Rong Yang, Danlin Li, Jie Hu, Run Tian, Yuhui Wan, Fangbiao Tao, Jun Fang, Shichen Zhang

**Affiliations:** 1Department of Maternal, Child and Adolescent Health, School of Public Health, Anhui Medical University, No 81 Meishan Road, Hefei 230032, China; yangrong@stu.ahmu.edu.cn (R.Y.); lidanlin@stu.ahmu.edu.cn (D.L.); hujie@stu.ahmu.edu.cn (J.H.); wyhayd@163.com (Y.W.);; 2MOE Key Laboratory of Population Health across Life Cycle/Anhui Provincial Key Laboratory of Population Health and Aristogenics, No 81 Meishan Road, Hefei 230032, Anhui, China; 3NHC Key Laboratory of Study on Abnormal Gametes and Reproductive Tract, No 81 Meishan Road, Hefei 230032, China; 4Department of Occupational and Environmental Health, School of Public Health, Guilin Medical University, No 1 Zhiyuan Road, Guilin 541199, China; tianrun@stu.glmc.edu.cn; 5Department of Toxicology, School of Public Health, Anhui Medical University, No 81 Meishan Road, Hefei 230032, China; 6Faculty of Pharmaceutical Science, Sojo University, Ikeda 4-22-1, Kumamoto 860-0082, Japan

**Keywords:** health literacy, health risk behaviors, latent class analysis, regression mixture modeling, Chinese adolescents

## Abstract

Adolescents engage in health risk behaviors (HRBs) that influence their current and future health status. Health literacy (HL) is defined as how well a person can get and understand the health information and services, and use them to make good health decisions. HL can be used to participate in everyday activities actively and apply new information to the changing circumstances. HRBs commonly co-occur in adolescence, and few researchers have examined how HL predicts multiple HRBs in adolescence. In this study we examined the subgroups of HRBs, and investigated heterogeneity in the effects of HL on the subgroups. In total, 22,628 middle school students (10,990 males and 11,638 females) in six cities were enrolled by multistage stratified cluster sampling from November 2015 to January 2016. The measurement of HL was based on the Chinese Adolescent Interactive Health Literacy Questionnaire (CAIHLQ). Analyses were conducted with regression mixture modeling approach (RMM) by Mplus. By this study we found four latent classes among Chinese adolescents: Low-risk class, moderate-risk class 1 (smoking/alcohol use (AU)/screen time (ST)), moderate-risk class 2 (non-suicidal self-injury (NSSI)/suicidal behaviors (SB)/unintentional injury (UI)), and high-risk class (smoking/AU/ST/NSSI/SB/UI) which were 64.0%, 4.5%, 28.8% and 2.7% of involved students, respectively. Negative correlations were found between HL and HRBs: higher HL accompanied decreased HBRs. Compared to the low-risk class, moderate-risk class 1 (smoking/AU/ST), moderate-risk class 2 (NSSI/SB/UI), and high-risk class (smoking/AU/ST/NSSI/SB/UI) showed OR (95%CI) values of 0.990 (0.982–0.998), 0.981 (0.979–0.983) and 0.965 (0.959–0.970), respectively. Moreover, there was heterogeneity in the profiles of HRBs and HL in different classes. It is important for practitioners to examine HRBs in multiple domains concurrently rather than individually in isolation. Interventions and research should not only target adolescents engaging in high levels of risky behavior but also adolescents who are engaging in lower levels of risky behavior.

## 1. Introduction

Health risk behaviors (HRBs) include specific forms of behavior, such as substance use, physical inactivity, non-suicidal self-injury (NSSI), suicidal behaviors (SB), unintentional injury (UI) and so on, which are proven to be associated with increased susceptibility to specific diseases or ill-health [1,2]. HRBs during adolescence are serious issues that contribute to significant physical, psychological and other consequences later in life.

The common substances abused by adolescents are tobacco and alcohol all over the world. The prevalence of current smoking habits among students aged 13–15 years in 61 countries ranges from 1.7% to 35.0% [3]. Worldwide, more than a quarter (26.5%) of all 15–19-year-olds are current drinkers, amounting to 155 million adolescents [4]. Even mild smoking behavior during adolescence was associated with an increased risk of regular smoking behavior in early adulthood [5]. Globally, overall 5.1% of the burden of disease and injury was attributable to alcohol [4]. In addition, tobacco and alcohol use (AU) in adolescence predicted lower educational achievement at later time [6]. Long-term cohort studies have shown that the more time spent on watching TV in childhood and adolescence, the greater the risk of overweight and obesity in adulthood [7]. Swannell et al. estimated that the international rate of NSSI among adolescents was 17.2% [8]. The prevalence of lifelong health problems in adolescents with NSSI was 17%–60% [9]. Globally, suicide is the second leading cause of death for people aged 15 to 29, and accidental injury has become the leading cause of death among teenagers aged between 10 and 19 years [10,11]. These results indicate that adolescent HRBs are prevalent and becoming a major public health problem. HRBs commonly co-occur as clusters, as people engage in multiple risk behaviors simultaneously [12]. For example, a number of evidences exist linking substance use, poor nutrition, physical inactivity, unhealthy weight control, and other HRBs among age groups [13,14,15]. HRBs commonly have a synergistic effect, so that the co-occurrence of multiple risk behaviors greatly increases the risk of chronic disease incidence and mortality, not only the additive effects of single behaviors [16,17,18]. A few studies have identified distinct class or clusters of HRBs which suggest that these classes vary among different groups of adolescents [19,20,21]. However, fewer studies have examined the subgroups and profiles of HRBs among Chinese adolescents based on nationwide data.

Among several factors that are related to HRBs in adolescents, health literacy (HL) has been identified as a modifiable factor which can be enhanced [22]. HL is defined as how well a person can get and understand health information and services, and use them to make good health decisions [23]. Furthermore, the field of HL includes three types: Functional literacy, interactive literacy and critical literacy. In 2008, Nutbeam proposed that HL is a more advanced cognitive and literacy skill which can be used to participate in everyday activities actively and apply new information to the changing circumstances. This definition emphasized that HL is made up of a set of skills [22]. The theoretical framework of adolescent HL proposed by Manganello suggests that HL could have different degrees of influence on a variety of health behaviors [24]. Research in health decision development suggests that there are usually two main models for adolescents to make decisions about health risks: Rational decision models (decisions based on perceived risk-reward tradeoffs) and reactive decision models (decisions based on immediate and direct judgments about the environment). The lower the HL level of adolescents, the worse their ability to make rational decisions, and thus making them more likely to make reactive decisions [25]. Actually, the negative association between HL and HRBs, such as smoking, AU, NSSI, UI, have been examined [26,27,28]. Nevertheless, most previous studies in this field have focused on its association with one or a few health behaviors. The traditional variable-oriented methods operate by partitioning variance between the dependent variable and changes in the independent variables, the results of this variable-oriented approach characterize the entire sample [29]. However, the conclusion of this method may be partially because the heterogeneity of HRBs has been identified in previous researches that one subgroup holds in common, but is different from other subgroups [19,20,21]. Therefore, we hypothesize that the association between HL and HRBs is different in these subgroups; the difference may be in direction, or it may be in magnitude. We plan to explore this conjecture in a person-centered method: Regression mixture modeling (RMM). Whereas both approaches use variables in the sense of manipulating quantitative data, person-centered strategies test for intraindividual and interindividual differences among the variables of interest. The results of person-oriented statistical methods are models of the relationships among the variable(s) of interest, which show the distinct configurations of heterogeneity within a sample. As far as we know, no study has explored potential heterogeneity in the regression of HL on subgroups of HRBs.

In consideration of the fact that long-term effects of HRBs on adolescents and decision-making skills in adolescents are not fully developed, it’s imperative to pay attention to adolescents’ HL and HRBs. However, relatively little research has examined how HL in adolescence can predict multiple HRBs. By investigating how HL predicts subgroups of HRBs in adolescent, the present study expands our understanding of the influence factors of HRBs. We expect that (1) potential subgroups of HRBs in Chinese adolescents will be identified, and (2) heterogeneity between HL and subgroups can be observed.

## 2. Methods

### 2.1. Study Participants and Procedures

This study was a cross-sectional study conducted in six cities in China and approved by the Ethics Committee of Anhui Medical University (1 March 2014; approval number 20140087). All subjects participated in the study upon receiving informed consent from their parents. Adolescent participants were recruited from junior and senior high schools by using convenient cluster sampling between November 2015 and January 2016. The participants were recruited from junior and senior high schools located in six cities in China, including both urban and rural regions, by using multistage stratified cluster sampling. Firstly, six cities were select by convenient sampling. These cities were Shenyang (capital of Liaoning Province), Xinxiang (North of Henan Province), Yangjiang (Southwest coast of Guangdong Province), Chongqing (one of China’s four direct-controlled municipalities), Ulanchap (Central Inner Mongolia Autonomous Region) and Bengbu (Northeastern of Anhui province). Then, eight schools (two rural junior and two senior schools, two urban junior and two senior schools) were selected in each region based on the stratified cluster sampling. Lastly, four to six classes were selected randomly from each grade in each school.

The investigator explained the purposes and procedures of the study to the students, and provided an opportunity for them to ask questions. The students were allowed to withdraw from the study if they were not willing to participate. Under the supervision of teachers, each participant completed a self-report questionnaire, including socio-demographic variables, HL, current smoking, current alcohol, ST, NSSI, SB, and UI, during 20–30 min in the classroom. A total of 23,137 students took part in this survey. Excluding the non-completion (missing data >5%) individuals, there were 22,628 valid questionnaires with the efficiency rate of 97.8%.

### 2.2. Measures

Socio-demographic variables were recorded as follows: age, gender (male or female), registered residence (urban or rural), household structure (only-child or more than one child), accommodation type (boarding student or commuting student), parental educational level (<high school degree or ≥high school degree), and self-reported family economic situation (bad, general, or good). Socio-demographic description of the sample was presented in Table 1.

The Chinese Adolescent Interactive Health Literacy Questionnaire (CAIHLQ) by Zhang et al. was used in this study and the reliability and validity have been demonstrated in previous study [30]. This scale assesses six dimensions (physical activities, interpersonal relationship, stress management, self-actualization, health awareness, dietary behavior) of HL with 31 items (e.g., ‘Follow a planned exercise program.’ ‘Take times with your family or friends.’ ‘Balance time between study and play.’ ‘Feel each day is very meaningful.’ ‘Containing sugars and food continuing sugar.’ ‘Eat 200–400 g of fresh fruit each day.’). Participants was asked to make a most suitable choice of each item from five answers (never and no desire, never but with desire, occasionally and irregularly, often, and routinely). In this study, internal consistency test showed that the Cronbach’s α was 0.910, and for each dimension Cronbach’s α was between 0.662 and 0.847. The total scores of each participant ranged from 31 to 155.

According to the Centers for Disease Control and Prevention definition of youth risk behavior surveillance system (YRBSS) [31], current smoking status was measured with the item “During the past 30 days, how many days did you smoke cigarettes? ” with response options of 0 = 0 days; 1 = 1 to 9 days; 2 = 10 to 19 days; 3 = 20 to 30 days, and current AU status was assessed with the item “During the past 30 days, on how many days did you have at least one drink of alcohol? ” with response options from 1 = 1 to 9 days to 3 = 20 to 30 days. The validity of self-reports current smoking and AU have been demonstrated [32,33].

All participants were asked about the average hours on weekdays spent on playing games or doing things unrelated to study on the computer every day (such as game console, music player, tablet computer, smartphone, QQ, Microblog, RenRen, and so on). According to the standard of American Academy of Pediatrics and previous studies, screen time (ST) >2 h/day is defined as too long ST [34,35,36].

NSSI of the participants over the previous 12 months before the survey were assessed by the Adolescent Non-suicidal Self-injury Assessment Questionnaire (ANSSIAQ), which included eight items (e.g., Have you ever hit yourself?) [34]. All the response options were “yes” or “no”. The details of the questions were as follows: (1) Have you ever hit yourself? (2) have you ever pulled your hair yourself? (3) have you ever banged your head or fist against something? (4) have you ever pinched or scratched yourself? (5) have you ever bitten yourself? (6) have you ever cut or pierced yourself? (7) have you ever exposed yourself to smoke, fire, and flames or come in contact with heat and hot substances? and (8) have you ever ingested a toxic substance or object? For those who confirmed that they had engaged in NSSI, the frequency of NSSI was investigated. As long as the answer was “yes” (one or more times), the student were judged as having NSSI behaviors. The reliability and validity have been demonstrated in a previous study [37]. The Cronbach’s α coefficient for NSSI in the present was 0.779, which was similar to previous research [38].

SBs were assessed as a composite of three item from YRBSS [31], being suicidal ideation (“Have you ever thought about killing yourself in the past 12 months?” with response options of 1 = none; 2 = once; 3 = 2 to 3 times; 4 = more than 4 times), suicide plan (Have you ever made a plan to kill yourself in the past 12 months?” with response options from 1 = none to 4 = more than 4 times), and suicide attempt (“Have you ever tried to kill yourself in the past 12 months?” with response options from 1 = none to 4 = more than 4 times). SBs were defined as a “yes” in response to any of the three questions. The validity of self-reported SB has been demonstrated [32,33].

According to the national standard of the People’s Republic of China [39] children and teenagers injury monitoring method, the UI are divided into road traffic incident, crush, falling and tripping, scratches, puncture or cut, bites and pricks, explosive impact, enclosed anoxic space, drowning, electric shock, chemical or other substances poisoning and other injuries. To evaluate the occurrence of accidental injury of the research objects in the past year, the statistical objects were judged as injury if once or more of the above situations occurred. The Cronbach’s α coefficient for UI in this study was 0.724. The validity of self-reported UI has been demonstrated [33]. The main measures in Table A1.

### 2.3. Statistical Analysis

Database was created by EpiData 3.1. Statistical analyses were performed with Mplus (Mplus Version 7.4) (Muthén & Muthén, Los Angeles, CA, USA) and SPSS 23.0 (SPSS Inc., Chicago, IL, USA). Measurement data is expressed as (mean ± SD). Normality test, population rate confidence interval was performed in SPSS software.

Regression mixture modeling (RMM) which is combined with latent class analysis (LCA) and traditional regression model is an approach to ascertain whether subgroups of individuals demonstrating differing relationships exist in the data. The basic assumption of LCA is that a few mutually exclusive latent classes can explain the probability distribution of diverse reactions of explicit variables, and each category has a specific tendency to select the response of each explicit variable [40]. Standard regression analysis assumes a homogeneous population and thus characterizes the relationship between predictors and outcomes using a single regression function. Rather, RMM searches for heterogeneity in the effects of a predictor on an outcome, which may vary in magnitude and/or direction, by building several regression equations [41]. In this study, we adopted a three-step approach for RMM modeling proposed by Vermunt [42]. Compared with the most likely class regression method [43], the most important feature of the three-step method is that the classification error or uncertainty is taken into account when the latent categories are grouped. The first step of the three-step estimation procedure requires estimating the latent class model. Six model fit indexes were adopted to help evaluating the optimal model when conducting LCA: Akaike Information Criterion (AIC), Bayesian Information Criterion (BIC), adjusted Bayesian Information Criterion (aBIC), Lo-Mendell-Rubin (LMR), Bootstrapped Likelihood Ratio Test (BLRT), and Entropy. LMR and BLRT are used to make a comparison between the estimated model and a model with *k*-1 class, or classes, with *k* equaling the number of classes [44]. For the LMR and BLRT, a low and significant *p* value signifies that the estimated model is superior to the model with one less class [44]. The AIC, BIC and aBIC are commonly used for comparing different counterpart models, with the lowest value on each indicator suggesting a best-fitting model [45]. Secondly, the procedure involves determining the measurement error for the most likely class variable; finally, the three-step procedure necessitates estimating the desired auxiliary model which the latent class variable is measured by the most likely class variable and the measurement error is fixed and prespecified to the values computed in step 2.

## 3. Results

### 3.1. Prevalence of HRBs

Of the 22,628 students, 10,990 were males (48.6%) and 11,638 were females (51.4%), and the age was 15.36 ± 1.79 years. The overall CAIHLQ mean score for all participants was 104.06 ± 18.68. A small minority (2.8%) of the sample reported smoking. The prevalence of AU was 16.8%. 16.3% of the respondents reported excessive ST. Approximately a third (32.1%) of the respondents reported having NSSI. The prevalence of SB was 15%. About half of the participants (46.7%) had an accidental injury.

### 3.2. Class Enumeration

Table 2 depicts the fit statistics for the one to five class models. The five-class model did not replicate the best log likelihood value and was therefore not considered further. The four-class solution was chosen as the final, best fitting model based on the lower AIC, as well as lowest BIC and aBIC values. Moreover, the *p* value of LMR was not statistically significant in class 5. With regard to model 4, bootstrap validation procedures also demonstrated that it had a good fit (*p* < 0.001). All remaining results are reported specific to the four-class solution.

As estimated from the model posterior probabilities, there were approximately 28.8%, 4.5%, 64.0%, and 2.7% of participants distributed across the four classes, respectively. There was generally good distinction among the four classes in this final model, based on an overall entropy value of 0.692, indicating good classification quality based on the threshold of 0.7 suggested by Nagin [46].

### 3.3. Characteristics of the Final Four Class Model

As shown in Figure 1, a low-risk class emerged (64.0%, *n* = 14,502), in which the prevalence of all kinds of HRBs was lower than other classes. In low-risk group, few students reported smoking (0.4%), AU (7.7%), ST (10.7%), NSSI (7.8%), and SB (6.1%); and over a third of these teens were also likely to have UI (35.4%). A high-risk class (smoking/AU/ST/NSSI/SB/UI) emerged (2.7%, *n* = 603), in which the majority of the adolescents engaged in smoking (22.3%), AU (76.5%), ST (58.6%), SB (52.5%) and UI (79.7%). In addition, all of them had NSSI in this class (100%). We found two moderate-risk classes, which we defined as those engaging in at least one or two of the negative behaviors but not all. The first moderate-risk class (4.5%, *n* = 1012) was characterized by fewer adolescent engaged in ST (49.1%), NSSI (18.9%), SB (20.0%) and UI (37.8%), with higher percentages of adolescents engaged in smoking (20.7%) and AU (72.2%). We refer to this class as the “smoking, AU, and ST” group. The second moderate-risk class (28.8%, *n* = 6511) was characterized by fewer adolescent engaged in smoking (1.5%), AU (16.9%), ST (16.0%) and SB (30.4%), but higher percentages of adolescents had NSSI (86.4%) and UI (72.5%). We refer to this class as the “NSSI, SB, and UI” class.

### 3.4. Effect of HL on the Best-Fitting Latent Classes of HRBs

Associations between the covariates and latent classes are presented in Table 3. One score of improvement in HL, as compared with the low-risk class, was significantly associated with a 1.0% lower risk among adolescents belonging to moderate-risk class 1 (smoking/AU/ST) (*OR* = 0.990, 95% *CI* = 0.982–0.998), a 1.9% lower risk among those in moderate-risk class 2 (NSSI/SB/UI) (*OR* = 0.981, 95% *CI* = 0.979–0.983), and a 3.5% lower relative risk among those with high risk (*OR* = 0.965, 95% *CI* = 0.959–0.970). The moderate-risk class 1 (smoking/AU/ST), compared to the low-risk class, were more likely to be older (*OR* = 1.327, 95% *CI* = 1.242–1.419), male (*OR* = 0.183, 95% *CI* = 0.137–0.245), commuting student (*OR* = 1.687, 95% *CI* = 1.323–2.151), and higher fathers’ educational level (high school or technical secondary school, junior college or above) (*OR* = 1.313, 95% *CI* = 1.035–1.664). The moderate-risk class 2 (NSSI/SB/UI), compared to the low-risk class, were more likely to be younger (*OR* = 0.838, 95% *CI* = 0.814–0.863), male (*OR* = 0.725, 95% *CI* = 0.662–0.795), urban (*OR* = 0.900, 95% *CI* = 0.816–0.993), commuting student (*OR* = 0.736, 95% *CI* = 0.665–0.815), and worse self-reported family economy (*OR* = 0.791, 95% *CI* = 0.715–0.874). The high-risk class (smoking/AU/ST/NSSI/SB/UI), compared to the low-risk class, were more likely to be male (*OR* = 0.359, 95% *CI* = 0.265–0.487), commuting student (*OR* = 1.636, 95% *CI* = 1.207–2.216), higher fathers’ educational level (high school or technical secondary school, junior college or above) (*OR* = 1.461, 95% *CI* = 1.084–1.968), and better self-reported family economy (*OR* = 1.752, 95% *CI* = 1.234–2.489). In summary, negative correlations were found between HL and HRBs: Higher HL accompanied with decreased HBRs. Males were at higher risk of various HRBs. The registered residence, accommodation type, father’s educational level, and self-reported family economy were the influencing factors of HRBs classification.

## 4. Discussion

The current study was designed to explore the latent classes of adolescent HRBs in a large Chinese sample. At the same time, as far as we know, this is the first study examined the association between HL and subgroups of HRBs by using RMM. This patterning helps to link a wide array of factors such as smoking, AU, ST, NSSI, SB and UI behaviors. Consistent with previous research examining multiple risk behaviors among adolescent [19,20,21], the present study found four categories, although the types and indicators of HRBs are not completely consistent and the age range or country of the respondents are different. Interestingly, HRBs appeared to show two different clustering patterns in the moderate-risk classes. The moderate-risk class 1 (smoking/AU/ST) had higher prevalence of smoking, AU, and ST. A systematic review found smoking and AU was the most commonly identified risk behavior cluster among adult [12]. This study shows that this clustering phenomenon also exists in adolescents, and adolescents with high prevalence of smoking and AU are also more likely to have ST for a long time. The moderate-risk class 2 (NSSI/SB/UI) had higher prevalence of NSSI, SB, and UI. This clustering pattern is not difficult to understand, because NSSI and SB often co-occur [47]. This study shows that this clustering phenomenon also exists in adolescents, and adolescents with a high prevalence of NSSI and SB are also more likely to have UI. A study reported a decrease in repetitive NSSI and SB in adolescents aged 15–17 in Germany at two-year follow-up, whereas high-risk substance misuse increased in this sample, especially in adolescents with frequent NSSI [48]. In addition, we found individuals were more likely to fall into the moderate-risk class 1 (smoking/AU/ST) as they age in our research. This could be an indication that in late adolescence, other dysfunctional behaviors will sometimes take over from NSSI. In designing health promotion strategies, it is important to understand the many challenges with which students are simultaneously struggling. For example, if most students with substance use are also experiencing high rates of too long ST, NSSI, and other HRBs, then narrow intervention approaches specifically targeting substance use behavior change (e.g., health education on smoking and drinking alcohol cessation) may not be effective in reaching students. Our results show that the proportion of the high-risk class (smoking/AU/ST/NSSI/SB/UI) is only 2.7%. However, given China’s population base, it is necessary to identify and intervene high-risk class (smoking/AU/ST/NSSI/SB/UI).

Some similarities were apparent across the moderate-risk class 1 (smoking/AU/ST), moderate-risk class 2 (NSSI/SB/UI) and high-risk class (smoking/AU/ST/NSSI/SB/UI) (smoking/AU/ST/NSSI/SB/UI). Relative to the low-risk group, all three subgroups were significantly more likely to be male and have lower HL scores. In the field of health and behavioral sciences, the study for differential effects of individual differences in the association between a predictors and outcome has become of increasing salience. It was implied in common theoretical viewpoints such as multifinality, which suggests that individuals exposed to similar conditions of risk may show different outcomes because of the heterogeneity of HRBs [49]. The present study verified the above viewpoint that although HL was associated with each subgroup, the strength of the associations was different. We found that HL was most associated with high-risk class (smoking/AU/ST/NSSI/SB/UI); when one score increased in HL, the probability of being admitted to the high-risk class (smoking/AU/ST/NSSI/SB/UI) was reduced by 3.5%. This suggests that we should pay attention to the HL level of students, especially in the high-risk group. In fact, changing HRBs by improving the HL level may be feasible. Levin-Zamir et al. found that media HL was positively related to a total health promoting behavior score [50]. Similarly, a research found functional HL was an independent predictor of a total health promoting behavior score [51]. As an individual ability, HL not only requires relevant health knowledge, but also requires the ability to analyze and make correct judgments independently. From a public health perspective, Nutbeam presented the concept of HL as an asset. HL is composed of a set of skills, and individuals can better manage their life events by effectively participating in social activities. At the same time, it is also implied that HL can be enhanced and strengthened through enhancing ability to reduce the occurrence of adverse health outcomes [22]. A large-scale cross-sectional study of 9–13 years old in 11 states showed that HL had a significant impact on certain health-related attitudes and ideas among adolescents. People who had difficulty understanding health information were not interested in or not willing to follow healthy educational content, and would also have an impact on the health management self-efficacy of children and adults [52,53].

This heterogeneity, which has the same direction but varies in magnitude, also appeared in gender; compared to the low-risk group, males were most likely to be classified as moderate-risk class 1 (smoking/AU/ST), followed by the high-risk class (smoking/AU/ST/NSSI/SB/UI), and moderate-risk class 2 (NSSI/SB/UI). Furthermore, regarding the low-risk group, there was heterogeneity in the relationship of registered residence, accommodation type, father’s educational level, and self-reported family economy to all three subgroups. Heterogeneity between demographic variables and subgroups of HRBs was also found in the study of Hair et al. [20]. Examining and clarifying cluster members demographic characteristics can help to establish intervention strategy by targeting those clusters precisely, which will greatly improve health promotion efforts. Namely, this study has significance for intervention research. In the past, most intervention programs were designed for certain risk behaviors or taking the scores of individuals on the scale as the criteria for intervention. Our results suggest that there was group heterogeneity in the occurrence of HRBs. In other words, different individuals showed different behavior types and also have certain homogeneity. It may be more effective to carry out interventions by targeting the subgroups of behaviors, rather than conventional intervention strategy focusing on certain risk behavior or all individuals as whole. Therefore, future intervention plans should be formulated according to the specific behavior model of each group, which could not only improve the efficacy of the intervention plan, but also enhance the work efficiency of practitioners and reduce the economic burden.

Although this study was a nationwide epidemiologic study with large samples, there are several limitations in the study. Firstly, as our analysis only utilized cross-sectional data, no conclusions about the causality of associations between HL and the latent classes can be determined. Secondly, not all kinds of HRBs were included. Although there are objective statistical indicators for the selection of the number of potential categories, there are still many subjective factors for the retention of the appropriate number, which results in differences among different studies and samples. Different samples may get different results, so the results of this study need to be verified by more studies in the future.

## 5. Conclusions

The potential impact of HL on HRBs is complex, but a better understanding of this relationship is useful from an HRB intervention standpoint. RMM was used to take a person-centered approach in characterizing this relationship, whereby individuals were classified into subgroups and the relationship between HL and subgroups was examined. The present study found four subgroups of HRBs among Chinese adolescents and HL could be an influencing factor for them. The proportion of low-risk class, moderate-risk class 1 (smoking/AU/ST), moderate-risk class 2 (NSSI/SB/UI), and high-risk class (smoking/AU/ST/NSSI/SB/UI) were 64.0%, 4.5%, 28.8% and 2.7%, respectively. While the heterogeneity of the association between HL and subgroups of HRBs showed HL was associated with each subgroup, the strength of the associations was different. We found that HL was most associated with high-risk class (smoking/AU/ST/NSSI/SB/UI). Future intervention plans should be formulated according to the specific behavior model of each group and HL should be promoted considering the profiles of HRBs. Further characterization of these subgroups using a variety of additional risk variables and outcome variables could give insights into potential constellations of risk behaviors among at-risk adolescents, which will also indicate avenues of future research in this area (i.e., the peer influence, the physical and psychological health, etc.) and provides direction for improving HRBs intervention development.

## Figures and Tables

**Figure 1 ijerph-16-03680-f001:**
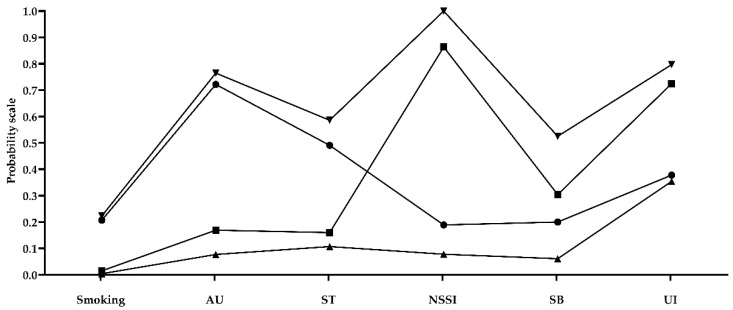
Four classes of health risk behaviors (HRBs) of the best-fitting four-class pattern. ▲ Low-risk class, 64.0%; ● Moderate-risk class 1 (smoking/AU/ST), 4.5%; ◼ Moderate-risk class 2 (NSSI/SB/UI), 28.8%; ▼ High-risk class, 2.7%.

**Table 1 ijerph-16-03680-t001:** Socio-demographic description of the sample.

Variable	Total Sample (%)
Gender	
Male	10,990 (48.6)
Female	11,638 (51.4)
Grade	
Middle school	11,993 (53.0)
High school	10,635 (47.0)
Registered residence	
Rural	10,882 (48.1)
Urban	11,746 (51.9)
Any siblings	
Yes	12,908 (57.0)
No	9720 (43.0)
Accommodation type	
Boarding student	11,320 (50.0)
Commuting student	11,308 (50.0)
Father’s educational level ^a^	
<High school degree	13,006 (57.5)
≥High school degree	9424 (41.6)
Mother’s educational level ^b^	
<High school degree	14,335 (63.4)
≥High school degree	8105 (35.8)
Self-reported family economy	
Bad	3240 (14.3)
General	16,345 (72.2)
Good	3043 (13.4)

^a^ 198 students have no father; ^b^ 188 students have no mother.

**Table 2 ijerph-16-03680-t002:** Model fit statistics for each of the fitted latent class analysis models.

Statistic	2 Classes	3 Classes	4 Classes	5 Classes
*df*	50	43	36	29
AIC	120,896.912	119,991.261	119,844.588	119,822.264
BIC	121,001.263	120,151.800	120,061.315	120,095.180
aBIC	120,959.949	120,088.241	119,975.510	119,987.129
LMR-LRT	<0.001	<0.001	<0.001	0.0592
BLRT	<0.001	<0.001	<0.001	<0.001
Entropy	0.549	0.725	0.692	0.579
Classification probability	0.28730 0.71270	0.24032 0.04698 0.71270	0.28774 0.04472 0.64089 0.02665	0.19299 0.28774 0.01003 0.47795 0.03129

*df*, degrees of freedom; AIC, Akaike Information Criteria; BIC, Bayesian Information Criteria; aBIC, Adjusted Bayesian Information Criteria; LMR-LRT, Lo–Mendell–Rubin Likelihood Ratio; BLRT, Bootstrapped Likelihood Ratio Tests.

**Table 3 ijerph-16-03680-t003:** Multinomial logistical regression predicting latent class membership.

Variable	Low-Risk Class	Moderate-Risk Class 1 (Smoking/AU/ST)	Moderate-Risk Class 2 (NSSI/SB/UI)	High-Risk Class (Smoking/AU/ST/NSSI/SB/UI)
	Adjusted *OR* (95% *CI*)	Adjusted *OR* (95% *CI*)	Adjusted *OR* (95% *CI*)	Adjusted *OR* (95% *CI*)
HL	ref.	0.990 (0.982–0.998) **	0.981 (0.979–0.983) ***	0.965 (0.959–0.970) ***
Age	ref.	1.327 (1.242–1.419) ***	0.838 (0.814–0.863) ***	0.978 (0.908–1.054)
Gender				
Male	ref.	ref.	ref.	ref.
Female	ref.	0.183 (0.137–0.245) ***	0.725 (0.662–0.795) ***	0.359 (0.265–0.487) ***
Registered residence				
Rural	ref.	ref.	ref.	ref.
Urban	ref.	1.176 (0.920–1.502)	0.900 (0.816–0.993) **	0.843 (0.629–1.129)
Household structure				
Only child	ref.	ref.	ref.	ref.
More than one child	ref.	0.940 (0.753–1.173)	0.957 (0.871–1.051)	1.157 (0.885–1.514)
Accommodation type				
Boarding student	ref.	ref.	ref.	ref.
Commuting student	ref.	1.687 (1.323–2.151)***	0.736 (0.665–0.815) ***	1.636 (1.207–2.216) **
Father’s educational level				
<High school degree	ref.	ref.	ref.	ref.
≥High school degree	ref.	1.313 (1.035–1.664)**	0.977 (0.874–1.093)	1.461 (1.084–1.968) **
Mother’s educational level				
< High school degree	ref.	ref.	ref.	ref.
≥ High school degree	ref.	0.945 (0.735–1.214)	0.994 (0.885–1.116)	0.951 (0.691–1.309)
Self-reported family economy (per level change)	ref.	1.015 (0.795–1.297)	0.791 (0.715–0.874) ***	1.752 (1.234–2.489) **

*OR* is odds ratio; *CI* is confidence interval; HL is health literacy; *** *p* < 0.001 compared with reference; ** *p* < 0.05 compared with reference. <High school degree included not graduated from primary school or attended school, primary school, and junior high school; ≥High school degree included high school or technical secondary school, junior college or above.

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
