# Peer review of "Association between Health Literacy and Subgroups of Health Risk Behaviors among Chinese Adolescents in Six Cities: A Study Using Regression Mixture Modeling"

_ijerph, 2019, doi:10.3390/ijerph16193680_

Round 1

Reviewer 1 Report

Dear Authors,

I read with a great interest your paper and I think that it is able to make an important contribution to the advancement of the scientific literature. However, I also found several spaces for improvement, which are summarized below in this revision letter. I will depict my main concerns with your research following the layout of the original version of your article.

ABSTRACT

The abstract is well written and it is compelling. However, after reading the abstract, it is not clear what do you mean for “health literacy” and how you assessed it. In my own opinion, you should improve the abstract, including some brief information about your own interpretation of HL and the approach you used to measure it. Moreover, you end up the abstract with the following statement “… The future intervention strategy of HRBs should be formulated according to the specific behavior model of each group and HL should be promoted considering the profiles of HRBs.” This statement is too generic and does not properly reflect the main implications of your research.

INTRODUCTION

The introduction shows several shortcomings. Firstly, it is not effective in setting the hook for the international reader. Actually, since the beginning of your article, the focus is put on China: however, although your research is rooted in the Chinese context, its implications can be useful at the international level. For this reason, you should make an effort to contextualize your research at the international (rather than at the local) level. Secondly, you do not show a clear and persuasive understanding of health literacy: since HL is one of the main concept triggering your research, you should show a thorough understanding of this construct, shedding light on its multifacetedness. Indeed, the scientific literature is not consistent in defining health literacy (please, refer to Pleasant et al., “Considerations for a new definition of health literacy”) and you should show a greater awareness of the state of the art in the field of health literacy. Thirdly, the introduction is muddled in several circumstances. As an illustrative example, it is not clear why –  at page 2, lines 84-95 – you report some methodological notes in the introduction. Fourthly – and lastly – the introduction is not effective in stating the main research questions this study is trying to give an answer to and, consequently, in stressing the originality of this article.

METHODS

More information is required about the “convenience” approach which was used to build the sample which participated in this research. Besides, the authors should better explain why the survey was filled under “…the supervision of teachers”: in fact, such a supervision may have biased the process of data collection. In section 2.2, the authors should include a table reporting the main socio-demographic attributes of the sample; similarly, in section 2.3, a table summarizing the main measures used in this study and some descriptive statistics for each measure will increase the clarity and the effectiveness of this paper.

RESULTS

Findings are well presented and effective. However, it is strange to read that “…approximately a third (32.1%) of the respondents reported having NSSI. The prevalence of suicidal behaviors was 15%”. It seems that the population investigated by the authors was highly problematic in terms of dangerous behaviors and this may have affected the consistency and the reliability of the research findings.

DISCUSSION

In the current version of the manuscript, the discussion section is primarily descriptive and it summarizes the main findings which were retrieved by the authors after data analysis. Alternatively, it only sporadically refers to the main conceptual and practical implications which can be obtained from the study findings. Moreover, in spite of the large sample involved in the analysis, it is quite ambitious to maintain that “…this study was a representative nationwide epidemiologic study”, due to the “convenience” approach used to build the sample.

CONCLUSION

The concluding section should be enhanced, pointing out the main take-aways of this article and illuminating the key policy insights that can be drawn from the research findings.

Once again, thank you very much for this interesting study. I hope that this revision letter will help you in further improving the quality of this intriguing and timely research.

Yours sincerely,

The reviewer

Reviewer 2 Report

The matter of the manuscript is interesting. The sample size is great.  The paper highlights the convenience of pay attention to the health literacy level of students, especially in high-risk group.

However, the manuscript has some weakness, which must be improved.

It has a good quality of statistical analysis although it is necessary to explain better the creation of the used questionnaire ( are all the questionnaire validated in this population?).  

It is not my expertise but sometimes it is difficult to understand some English sentences. Please review the English style in the manuscript by a native speaker.  Example p.85 ” An person-centered approach to solving this question is more deductive and data-driven, which   seeks to understand whether there are particular subgroups in a population for whom a  relationship exists”. p. 108” All  subjects participated in the study upon receiving informed consent form their parents”. p.270 “this is the first study examined the association  between HL and latent classes of HRBs by using RMM”.

P.73 I recommend, defining HL in the text the first time, independently of the abstract.

P140. “According to the WHO definition of youth risk behavior surveillance system (YRBSS), Please cite the reference or insert one.

P148 “According to the standard of American Academy of Pediatrics, screen time >2 hours/day is defined as too long  screen time”. Please cite the report or paper. I think it could oscillate in function of the age of the minor or adolescent.

P238. “Some similarities were apparent across the moderate-risk class 1 (smoking/drinking/screen 283 time), moderate-risk class 2 (NSSI/Suicidal behaviors/unintentional injurious) and high-risk class.” Please do the same with the high-risk class. Introduce it in brackets.

p.304” People who had difficulty understanding health information were not interested in or not willing to follow healthy educational  content, and would also have an impact on the health management self-efficacy of children and  adults [45, 46].” Unintelligible.

p.308-325. Discussion: Please improve and clarify the discussion. The writing style must be improved. Some sentences are difficult to understand. i.e.(p321): “If interventions are not targeted at individuals with specific types of behavior, behavior changes in certain categories of individuals will be delayed”.

Along the text, there is a wide use of acronyms. I recommend inserting a glossary of terminology at the end of the manuscript. Example: health risk behaviours, health literacy, etc.

Reviewer 3 Report

The study was conducted in very large sample among Chinese teenagers. . This study investigates how HL predict of HRBs in Chinese adolescents, and by using RMM try to define its’ influence to these behaviors. The method of Regression Mixture Modeling (RMM) has been applied to examine the relationships between HL and subgroups of individuals demonstrating different level of HRBs classes. Identification of these subgroups of adolescents at-risk gives better understanding of the determinants. And this allows to plan more effective adolescents’ health promotion and health education programs.

In the Introduction part the literature study is abundant and elaborated in synthetic way.

In Methods I do not see need to separate the subchapter 2.2. Demographic information, and I would suggest to give more general subtitle as “Study Design” and “Measures” or so.

Line 140: the reference to the WHO is not correct. The YRBSS is published by the DC (Centers for Disease Control and Prevention). There should be included the right reference to CDC publication.

Line 149: Reference to the standards of “screen time” defined by the American Academy of Pediatrics should be included too.

Line 224-24: The explanation, description of high-risk class is missed in the text, and should be written up.

Line 246-265: There is lack of interpretation of the results shown in Table 2 regarding father’s and mother’s school degree. This should be completed, fill in the text. The final sentences of the summary of the results shown in the table 2 is missed too.

English corrections :

Please correct “injurious” for “injuries”

“Drinking” should be corrected as “drinking alcohol” or “alcohol use”

Small corrections required in English in: Line 267

Line 288-290: Sentence is not clear, not understandable, should be re-written.

In the References please correct position 12.

Round 2

Reviewer 1 Report

Dear Authors,

Thank you for this revised version of your work. I now think that the paper is ready for publication.

Once again, well done with this interesting research.

Yours,

The reviewer